# Breaking Barriers to Rapid Whole Genome Sequencing in Pediatrics: Michigan’s Project Baby Deer

**DOI:** 10.3390/children10010106

**Published:** 2023-01-04

**Authors:** Caleb P. Bupp, Elizabeth G. Ames, Madison K. Arenchild, Sara Caylor, David P. Dimmock, Joseph D. Fakhoury, Padmani Karna, April Lehman, Cristian I. Meghea, Vinod Misra, Danielle A. Nolan, Jessica O’Shea, Aditi Sharangpani, Linda S. Franck, Andrea Scheurer-Monaghan

**Affiliations:** 1Corewell Health Helen DeVos Children’s Hospital, Grand Rapids, MI 49503, USA; 2Department of Pediatrics and Human Development, College of Human Medicine, Michigan State University, Grand Rapids, MI 49503, USA; 3Sparrow Hospital, University of Michigan Health System, Lansing, MI 48912, USA; 4Rady Children’s Institute for Genomic Medicine, San Diego, CA 92123, USA; 5Pediatric Hospital Medicine, Bronson Children’s Hospital, Kalamazoo, MI 49007, USA; 6Department of Pediatric and Adolescent Medicine, Homer Stryker School of Medicine, Western Michigan University, Kalamazoo, MI 49007, USA; 7Department of Obstetrics, Gynecology and Reproductive Biology, College of Human Medicine, Michigan State University, Lansing, MI 48912, USA; 8Children’s Hospital of Michigan, Central Michigan University, Detroit, MI 48201, USA; 9Beaumont Children’s Hospital, Detroit, MI 48073, USA; 10Department of Family Health Care Nursing, University of California, San Francisco, CA 94143, USA; 11Neonatal Intensive Care, Bronson Children’s Hospital, Kalamazoo, MI 49007, USA

**Keywords:** rapid whole genome sequencing, genetics, genomics, pediatrics, hospital medicine, medical technology and advancement, reimbursement, quality improvement

## Abstract

The integration of precision medicine in the care of hospitalized children is ever evolving. However, access to new genomic diagnostics such as rapid whole genome sequencing (rWGS) is hindered by barriers in implementation. Michigan’s Project Baby Deer (PBD) is a multi-center collaborative effort that sought to break down barriers to access by offering rWGS to critically ill neonatal and pediatric inpatients in Michigan. The clinical champion team used a standardized approach with inclusion and exclusion criteria, shared learning, and quality improvement evaluation of the project’s impact on the clinical outcomes and economics of inpatient rWGS. Hospitals, including those without on-site geneticists or genetic counselors, noted positive clinical impacts, accelerating time to definitive treatment for project patients. Between 95–214 hospital days were avoided, net savings of $4155 per patient, and family experience of care was improved. The project spurred policy advancement when Michigan became the first state in the United States to have a Medicaid policy with carve-out payment to hospitals for rWGS testing. This state project demonstrates how front-line clinician champions can directly improve access to new technology for pediatric patients and serves as a roadmap for expanding clinical implementation of evidence-based precision medicine technologies.

## 1. Introduction

The use of rapid whole genome sequencing (rWGS) early in the evaluation of hospitalized infants and children with critical illness of unclear etiology has reproducibly demonstrated improved diagnostic yield, faster time to diagnosis, reduced costs of care, enhanced physician and parental satisfaction, and improved patient outcomes [1,2,3,4,5,6,7,8,9]. However, there are often significant barriers to implementation of new technologies in healthcare [10]. A previous quality improvement project, Project Baby Bear (PBB), provided access to rWGS for critically ill neonates at five medical centers in California. Qualitative analysis of the implementation process revealed five key themes: the need for rWGS champions; clinician educational needs and strategies; negotiating decision-making roles and processes; workflows and workarounds; and impact of healthcare worker perceptions about rWGS [11].

Project Baby Bear marked the beginning of a transition to clinical use of rWGS as a cost-effective precision medicine tool within routine practice. The success of PBB inspired clinicians at children’s hospitals in Michigan to formalize a coalition to advance the adoption of rWGS technology to better serve acutely ill infants and children in Michigan. In 2018, Helen DeVos Children’s Hospital (HDVCH) became the first hospital in Michigan to make rWGS accessible for clinical use, and in early 2019, Bronson Methodist Hospital (BMH) also began clinical use of rWGS. Concerns about access to testing across the state, costs of supporting testing, lack of access to genetics professionals and limited knowledge among clinicians at other children’s hospitals in Michigan, drove further collaboration. This paper describes the processes by which frontline clinicians worked with the state hospital association and other stakeholders to address barriers to access to rWGS for children with critical illness across Michigan. Lessons learned from this collaboration can inform healthcare professionals in other regions who seek to improve clinical care through increased access to rWGS.

## 2. Materials and Methods

The project occurred in three phases: exploration, preparation, and implementation. Evaluation metrics were identified and incorporated as the project evolved. Each of the phases is described below.

### 2.1. Exploration

In the exploration phase, clinical teams from the early adopting hospitals, HDVCH and BMH, connected with the Michigan Health and Hospital Association (MHA) to discover ways to expand access to rWGS across the state. The MHA leaders suggested presenting the idea of a statewide rWGS initiative to their Children’s Health Council, a group composed of executive representatives from the Michigan hospitals with pediatric inpatient services. In the spring of 2019, a medical geneticist from HDVCH introduced rWGS to the MHA Children’s Health Council, and a neonatologist from BMH shared clinical and economic outcome data from PBB. Michigan Health and Hospital Association Children’s Health Council representatives and MHA leaders expressed overwhelming support, and a commitment was made to pursue an rWGS implementation program in Michigan, named Project Baby Deer (PBD).

Following this, two key MHA representatives visited the Rady Children’s Institute for Genomic Medicine (RCIGM) in California to gather insight about rWGS and PBB. To broadly share the idea of a statewide rWGS project, MHA invited key stakeholders from the 14 Michigan hospitals with level III or IV neonatal intensive care units (NICU) for an initial informational meeting. In November 2019, under the auspices of the MHA in Okemos, MI, a broad coalition of interested parties from 11 hospitals gathered to evaluate the potential use of rWGS for critically ill neonatal and pediatric inpatients in Michigan. Attendees included physician and administrative leaders, experts in health policy and advocacy from MHA, and outcomes researchers and clinicians from RCIGM. The aim was to offer rWGS to critically ill neonatal and pediatric inpatients at Michigan hospitals across the state using a standardized approach that included: (1) agreed inclusion and exclusion criteria for eligible cases; (2) pooled funding; (3) ongoing collaboration and shared learning; (4) standardized health economic impact evaluation metrics; (5) considering immediate and long-term reimbursement strategies; and (6) engagement in state and federal advocacy.

Agreements were established for RCIGM to perform rWGS on samples sent from the participating Michigan hospitals. Procedures for rWGS ordering, sample acquisition and results reporting were performed accordingly to previously published workflows, with a preference for trio testing when ultra-rapid whole genome sequencing was requested. This included clinical features and severity of illness driving the decision to order testing, blood samples being used to obtain DNA for sequencing, and results being disclosed from the testing laboratory to ordering provider [12]. To reduce barriers to patient access to testing, PBD included an agreement that in the initial phase of the project the cost of the sequencing, interpretation, and analysis of genomics would be funded by a generous grant from the J. Willard and Alice S. Marriott Foundation. Additional funding for rWGS included the RCIGM rWGS Donor Restricted Fund, and an additional grant from the Michigan-based Children’s Foundation. Some Michigan hospitals provided support for the uncovered costs of testing from their individual sites. This funding enabled broad access to rWGS for all patients meeting agreed inclusion criteria and allowed PBD champions to focus on the other aspects of rWGS implementation such as education and advocacy.

### 2.2. Preparation

As the project transitioned from exploration to preparation for implementation, group norms were established, and relationships formalized including monthly PBD meetings organized by MHA and RCIGM, with attendance from Michigan-based geneticists and neonatologists as well as policy and advocacy experts. This collaborative group agreed on inclusion and exclusion criteria and to minimum genome-wide sequencing standards for all cases in the project (Table 1).

Multi-site collaborative agreements across institutions were required prior to application of rWGS in clinical practice at each site. First, physician champions and hospital administrative champions were identified at all sites. The physician champions were responsible for ensuring frontline clinical teams were engaged and informed about the project while encouraging appropriate clinical identification of cases and effective internal communication. Administrative champions were responsible for executing the project commitment form, establishing a lab service agreement with at least one lab able to perform rWGS, ensuring research ethics review by local institutional review boards (IRB), and working with the physician champion(s) to create a project onboarding timeline (Figure 1; Table 2). All participating institution IRBs deemed this work quality improvement and therefore exempt from IRB oversight.

Project set-up at each site was either in-person or virtual and included individual and multi-site onboarding sessions to educate clinicians on using the electronic portal for ordering rWGS and uploading clinical notes to guide analysis of the genome. This could include pertinent clinical information from the ordering provider and Human Phenotype Ontology (HPO) terms, genetic conditions or genes high on differential diagnosis, or relevant imaging and lab results to the testing lab to aid in matching gene variants identified with potential genetic diseases. Education for clinicians included review of the rWGS protocol, testing specifics, patient eligibility, consent, sample acquisition and phenotypic data transfer. Resource materials were provided for testing consent, parent counseling, and documentation. A PBD toolkit with onboarding documents, workflows, technical and educational information about the testing, parent handouts, and contact lists were mailed to each participating hospital. Education about genomic medicine, genetic testing, and rWGS was provided at no charge to PBD sites, through a “Genomics 101” course at the 5th Annual Frontiers in Pediatric Genomic Medicine Conference in April 2020 [13].

### 2.3. Implementation

The implementation phase focused on integrating the rWGS protocol into clinical care at each site. The protocol covered: case identification, rWGS ordering and portal usage, sample and phenotype information transmission, establishing a pathway for rapid return of results to clinicians for use in clinical decision-making, and distribution of a clinical impact survey to collect and review resultant changes in care using REDCap electronic data capture tools.

A monthly virtual statewide PBD case review conference was implemented and open to all participating sites as well as interested non-participating sites. Medical directors from Michigan-based payors, commercial and Medicaid, were also invited to attend this conference. These conferences included de-identified presentations of 1–2 clinical cases for which rWGS was used from participating Michigan hospital teams to guide diagnosis and medical management of patients. In the monthly case reviews, presenters shared the clinical scenario, described their decision-making process, reviewed how rWGS was utilized in the case, discussed outcomes of testing and integration into care, and discussed lessons learned from the case. As rWGS was utilized throughout the state, informal conversations and discussions evolved among sites, particularly between those with and without clinical genetics teams, to support each other as testing was new to many centers. In addition, some centers developed new formal telemedicine partnerships to provide virtual assistance from genetics professionals.

In practice all testing was performed at RCIGM following methods as previously described [12]. Briefly PCR free whole genome data was generated on an Illumina NovaSeq 6000 to a minimum 40× depth. Variant calling including copy number variants was undertaken using Dragen. Variants were manually interpreted iteratively by clinical molecular geneticists according to ACMG reporting guidelines with only “Pathogenic”, “Likely Pathogenic” and ‘Suspicious Variants of Uncertain Significance (VUSS)’ returned to clinical sites.

During implementation, PBD champions also prioritized payor advocacy with the goal of establishing insurance coverage for any infant or child in need of rWGS. The experienced state advocacy leaders at MHA successfully engaged with Michigan Medicaid and Blue Cross Blue Shield of Michigan. For example, PBD champions presented at the Michigan Association of Health Plans Medical Director meeting a sample clinical case involving rWGS along with brief review of California’s PBB data and the PBD proposal. This discussion helped raise awareness and improve engagement with in-state payors early in the project.

The PBD team also evaluated health care worker knowledge and attitudes toward rWGS and implementation of rWGS in clinical practice at PBD sites. A standardized survey was distributed by email to healthcare professionals at nine hospital sites in Michigan. Information was gathered about capacity to implement rWGS as well as personal knowledge, education, and attitudes about future utilization of rWGS and genomics [14]. This work informed ongoing educational and quality improvement activities.

## 3. Results

As of May 2022, there were seven on-boarded PBD hospitals across Michigan (Figure 2). Some barriers to implementation were seen at all sites, including those with and without medical geneticists and genetic counselors. Logistical barriers included constrained bandwidth of clinical champions, challenges with institutional review boards understanding the distinction between research and quality improvement, concern about data storage and consent procedures, alternate preferred labs not initially being able to meet expected turnaround times, and difficulty or delay in executing lab service agreements to enable send-out of samples to RCIGM for rWGS.

### 3.1. Clinical Impact

The clinical impact of rWGS was evaluated by clinician assessment of change in management (Appendix A). The primary treating clinician involved in the patient’s care when rWGS results were returned was asked to complete the assessment after each result was returned. The individual case assessment surveys were analyzed using descriptive statistics. A PBD summary document highlighting clinical and economic impacts was created in February 2021, with the first 30 children sequenced through the program. This document was revised in March 2022 with the addition of subsequent cases and distributed to participating hospitals [15].

As of November 2021, 18 months after the PBD launch, a total of 89 infants and children had received rWGS through the project. Completed clinical impact surveys were available from 64 of the 89 cases (72%). Analysis revealed a diagnostic rate of 39% and change in management rate of 27% (Table 3). Among the 24 patients for whom a significant change in care was documented, there were between 95 and 214 inpatient days avoided, multiple avoided surgeries (e.g., lung biopsy, tracheostomy, muscle biopsy, and skin biopsy), appropriate medications prescribed, and initiation of evaluation for a heart transplant.

Notably, to avoid over-estimating benefit, missing clinical impact surveys were conservatively interpreted as having no significant change in patient management. Consistent with other studies such as PBB and NICUSeq, slightly more males (64%) than females were tested [2,4]. Race and ethnicity were reported in 94% and 99% of the cases, respectively and is representative of the Michigan population (Appendix A) [16].

In 62 of the 89 cases sequencing was of proband only, 3 were duo with one parent included, and 24 were trio. In this cohort there was not a major difference observed in diagnostic rate between proband-only cases (40% diagnostic) and trio cases (42%). The similarities in diagnostic rate are expected in this sample size as previous published [6].

### 3.2. Economic Impact

An economic impact analysis was performed using data from the clinical impact surveys. The economic impact analysis followed the methodology of PBB to assess monetary savings by calculating the avoided healthcare costs resulting from decreased length of stay, avoided tests and procedures, and reduced professional fees associated with the shorter length of stay in the ICU [17]. This analysis uses an average savings associated with an inpatient day avoided calculated from the PBB Medicaid data as no such data was available from Michigan Medicaid specifically. The cost of rWGS was calculated using an average cost per case at the Michigan Medicaid established rates for covered Current Procedural Terminology (CPT) codes [18] and the number of proband and comparator genomes sequenced in the PBD cohort. The gross savings were calculated as the midpoint of the estimated range of avoided days multiplied by the average savings associated with an avoided inpatient day. The net benefit was calculated as the gross savings less the cost of rWGS (average cost per case multiplied by volume of cases). The net benefit per patient was defined as the net benefit divided by the volume of contributing cases. With an average savings per inpatient day avoided of $4854.47 and average cost per case of $7563.86, the analysis estimated a net saving per patient of $4155.13 (Table 4). Under this model assumptions, the results of this economic impact analysis demonstrate a positive net benefit per patient when rWGS is provided as a first-tier test to clinically qualified patients in the public payer system. Expecting that commercial payers reimburse inpatient care at a higher rate than Medicaid, it is likely the net benefit in a commercially insured or mixed payer population would be greater.

### 3.3. Family Impact

Clinical champions informally shared favorable feedback from families regarding the use of rWGS in diagnosis and care management for their children, both from families who received a diagnosis and those who did not. Some families expressed the wish that rWGS had been done sooner in the hospital stay. Many families felt that all children should have access to quick answers and early prevention to enable the most optimal outcomes. Telemedicine support for genetic counseling was successful, one mother commenting that “We felt very supported by the genetics department from afar”. The perspectives of four families who participated in PBD were shared in the March 2022 project summary document.

### 3.4. Policy Impact

Advocacy efforts led to a rewarding working relationship between PBD clinical champions and leaders from the Michigan Department of Health and Human Services (MDHHS). This relationship was rooted in the mutual goal of establishing a coverage policy for rWGS with consideration for accessibility, inclusivity, clinical appropriateness, and utilization management. PBD champions were gaining experience in rWGS utilization at the time of policy development and were able, in real time, to effectively offer input on policy content. This collaboration served to strengthen the final policy and allow for meaningful change in access aligned with current clinician practice. This positive relationship cultivated a sense of trust and partnership within the PBD and MDHHS team. Clinical champions, stakeholders and collaborators achieved advocacy success when the Michigan rWGS Medicaid policy went live on September 1, 2021, making Michigan the first state in the nation to have a carve-out payment to the ordering hospital for rWGS [19].

## 4. Discussion

The goals of PBD were to address barriers to protocol-driven equitable access to rWGS in clinical patient care for acutely ill neonatal and pediatric patients. The project also sought to create model systems for implementation, quality improvement monitoring, and dissemination of learnings. These goals were achieved because of a clinician-driven multi-stakeholder collaborative effort. Key factors contributing to the success of Michigan’s PBD included project philanthropic funding as well as support from the state hospital association, hospital executives, administrators, frontline clinicians, and genomics experts. Local philanthropic support allowed two children’s hospitals in Michigan to become early adopters of rWGS and the resultant positive experiences inspired a goal to offer access to this precision medicine diagnostic for clinicians caring for ill infants or children with uncertain diagnoses regardless of location in the state or type of insurance.

The partnerships with MHA and RCIGM were key to success of PBD. The partnership with MHA provided crucial statewide connections to hospitals, healthcare systems, and payors. The work of PBD was aligned with the mission, vision, and values of MHA’s Keystone Center, which seeks improvements in safety and quality outcomes in healthcare, providing an important cornerstone of a strong collaborative network that was necessary to achieve success [20]. Rady Children’s Institute for Genomic Medicine provided scientific expertise, laboratory capacity, administrative support, some funding for initial rWGS, and implementation experience from PBB.

As has been found in previous research, the role of the frontline clinical champion is essential to successful implementation of rWGS in clinical care. This collaborative group of dedicated individuals was unified in their commitment to bring rWGS to their hospitals as a powerful tool, proven to improve clinical outcomes for infants and children, reduce the costs of care, and improve the experience of care for families and providers. Administrative and executive support from key hospital systems and pediatric centers were also important for developing a collegial network sharing a commitment to improve care for children. In this way, the group was able to avoid some of the competitive barriers that can stand in the way of statewide partnerships and projects like PBD. The champions provide leadership at all levels including engaging clinical colleagues in developing guidelines and using rWGS effectively in clinical care, identifying need for further education or addressing logistical barriers, and in advocacy at the hospital and state level to address access and health equity. Diverse clinical representation amongst clinical champions was also important to allow appropriate expansion of the patient population receiving testing. As an example, support from pediatric hospital medicine champions clarified the potential usage and benefit of rWGS outside the intensive care unit and served as a key liaison to other inpatient providers. Not every participating site had in-person or virtual access to genetics physicians and genetic counselors. During PBD, new collaborations between hospitals began and evolved to increase that access.

The 39% rWGS diagnostic rate and the change in management rate of 27% found in the first 18 months of the project are consistent with previous literature. As more information becomes available regarding the clinical and economic benefits of rWGS in pediatric clinical practice, focus shifts to implementation across pediatric inpatient settings at scale. California’s PBB is an example of preliminary scale-up of rWGS utilization at a state level. The 100,000 Genomes Project in the United Kingdom through National Health Service England is an example of country-wide scale-up of clinical genomic testing [21]. In the fragmented United States (US) healthcare system, children who may benefit from rWGS are not covered by the same insurance, with differences in payor approaches to genetic testing and reimbursement, including differences between state Medicaid programs for publicly insured patients. A significant success of PBD is the 2021 Michigan Medicaid coverage and reimbursement policy. It was recognized early on that payor coverage for rWGS needed to be a carve-out type of reimbursement. The cost of rWGS as a part of the standard hospital inpatient diagnosis-related group billing system disincentivizes utilization of higher cost testing like rWGS. This causes rWGS to be underutilized despite multiple studies demonstrating cost savings associated with rWGS and no other inpatient testing being held to such a high standard as genetic testing. In the current US healthcare payor system, Medicaid partnership and support is critical to successful implementation of new technologies, and PBD emphasized the importance of developing those relationships and maintaining dialogue with key Michigan Medicaid leaders. Coverage policies for rWGS are subsequently being implemented in other states including California [22], Minnesota [23], Louisiana [24], Maryland [25], and Oregon [26].

While the Michigan Medicaid rWGS policy was the first of its kind, the coverage is for inpatients under one year of age only. Access to testing for children in the outpatient setting and at ages beyond this limit remains a challenge. Researchers recently reported that infants who undergo genome-wide sequencing continue to accrue cost benefit over their lifetime, up to $18,877 per quality-adjusted life year [1]. Ongoing evaluation of the clinical and cost benefits of rWGS will be needed, particularly as policies in other states have differences in their terms and limitations.

Genomic disease affects individuals of all races and ethnicities, thus there is need to ensure that any pediatric patient not only has access to testing, but early access that is independent of location or insurer. Equity in access to rWGS does not translate to equity in care delivery after diagnosis, and there is a continued need to ensure families receive treatments and follow up in an equitable manner [27,28]. The geographic locations of inpatient medical centers that care for children are typically unevenly spread, even considering population. Michigan is the 10th largest by population, and 22nd largest state in the US by square mileage, yet the availability of local pediatric inpatient care can still be hours away for a child and family in need (Figure 2). Equity in access to hospitals with pediatric subspecialty expertise remains a challenge for many children and their families. The availability of genetics professionals is limited and tends to be concentrated at a limited number of sites within states. Access may improve with expansion of virtual genetics consultation. A successful example of telemedicine genetic support now exists between HDVCH and BMH. Knowledge about and comfort in utilizing rWGS is also limited among pediatric clinicians generally. There are important ethical concerns that pertain to rWGS which may impact utilization and further study of parental experience with rWGS is needed [29,30]. Each of these factors creates an uneven landscape upon which to implement and optimize genomic sequencing informed precision medicine for neonatal and pediatric patients.

With the success of Michigan’s PBD thus far, future potential and plans abound. Providing access to rWGS and appropriate follow up care remains a priority to the PBD team. Perhaps the greatest challenge going forward is prioritizing opportunities and maintaining stakeholder engagement and momentum. Barriers to rWGS access persist at some Michigan sites due to lack of clinical champions, resources, clinician confidence to implement rWGS, as well as legal and contracting deficiencies. Even existing and engaged clinical champions will need more dedicated time and teamwork to maximize the potential of rWGS at their centers and to implement quality improvement monitoring to ensure appropriate utilization.

There is a clear need for provision of education around genetic concepts and genomic testing. This includes clinicians, genetic and non-genetic, as well as other members of the healthcare team such as nurses, social workers, and administrators. The PBD team is currently developing education initiatives to increase healthcare worker interest, knowledge, and comfort with rWGS. Presentation and discussion of PBD has also happened at various conferences and meetings include the Michigan Chapter of the American Academy of Pediatrics and Society of Michigan Neonatologists annual meetings. With more states beginning coverage policies for rWGS, continued discussion and dialogue are needed to learn from expanded usage.

Billing and reimbursement workflows also need to be developed and standardized. Due to the lack of experience with seeking approval and payment for the carve-out nature of inpatient billing, there is a need for ongoing input from lab managers, clinicians, and state billing representatives to optimize appropriate testing approval. These processes also differ between individual medical centers where personnel, billing workflows, and methods of initiating policies and procedures vary. Even with the clinical benefits of rWGS and the ability to have the cost of testing covered, if the processes involved are too cumbersome for providers and others involved, utilization will not be optimal. There is a need for ongoing improvement and collaboration in this area.

Limitations to the current rWGS coverage in Michigan deserve mention. As an example, Medicaid coverage stops at one year of age, and multiple PBD patients that received meaningful rWGS results were well over that age. Commercial payors in Michigan have also begun to explore coverage policies, though not all with carve-out payment. These policies have differing inclusion and exclusion criteria, cut-off ages, and methods for submitting for approval and potential reimbursement. As rWGS usage increases, the challenge of determining meaningful inclusion and exclusion criteria also evolves. For example, hypoxic-ischemic encephalopathy with clear precipitating event is often on the list of exclusion criteria for rWGS. However, as the spectrum of pediatric genomic disease is further elucidated, ongoing exclusion of these patients from inpatient rWGS risks missing identification of potentially relevant clinical information such as a genetic predisposition to seizures. Lastly, the proposition of the economic benefits of rWGS should be considered in value-based care and risk-sharing agreements within healthcare, as those create significant opportunity to leverage the impact on the cost-of-care on top of the well-established clinical benefits of rWGS. While patients in the PBD cohort included both publicly and privately insured patients, this analysis used Medicaid reimbursement rates because there is more robust data available from Medicaid to compare estimated costs and benefits and reduce assumptions as rates from commercial payers may be highly varied among sites and payers. Ongoing efforts to expand and improve payor coverage will be essential, both in expanding criteria appropriately and equitably, but also seeking engagement with payors in Michigan not yet involved.

Continued assessment of clinical, economic, and social outcomes over time is also important, as a better understanding of this longer-term impact of rWGS should help inform appropriate expansion of rWGS inclusion criteria with the goal of providing rapid diagnosis and improved care for more children. Gathering clinical data, results, outcomes, and economic impact in a functional repository for further study would also help improve many processes. Family engagement to understand the patient experience of rWGS also needs to be formally investigated. Telemedicine relationships between centers with and without onsite geneticists are likely to help combat difficulties with access to and follow up with genetic teams. Lastly, helping other states develop similar initiatives to PBD will be useful to continue to define and refine the landscape of genome sequencing informed precision medicine for neonatal and pediatric patients.

## 5. Conclusions

Perhaps of greatest importance to PBD’s success was the sheer patience and perseverance of the PBD champions. The process and experience described here was a ‘grass root’ effort starting simply with one person talking to another person. As those conversations expanded and multiplied, consistent effort was needed to persevere through each small, and often time-consuming step. Sustaining this work will take continued engagement education of providers and hospitals, increased access to genetic testing, improvement in costs and payor acceptance and expansion of experience to other states and countries. PBD is an example of a successful model for rWGS implementation, providing an approach for other states to follow and refine, as well as providing a microcosm of what could be possible on a federal level.

## Figures and Tables

**Figure 1 children-10-00106-f001:**
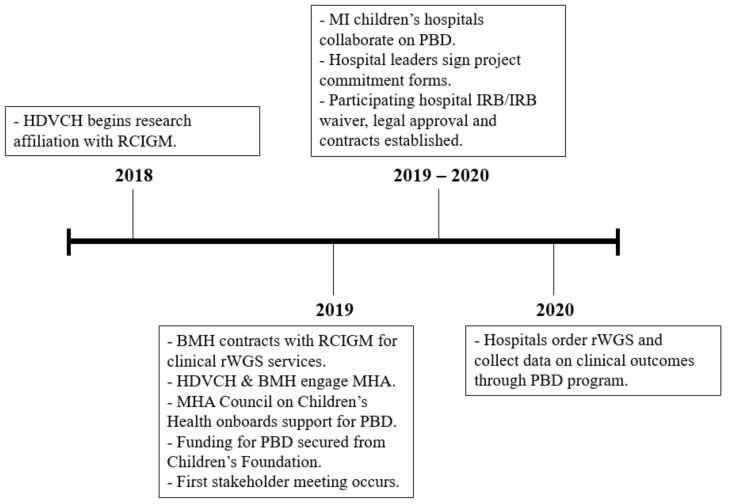
Timeline for Project Baby Deer.

**Figure 2 children-10-00106-f002:**
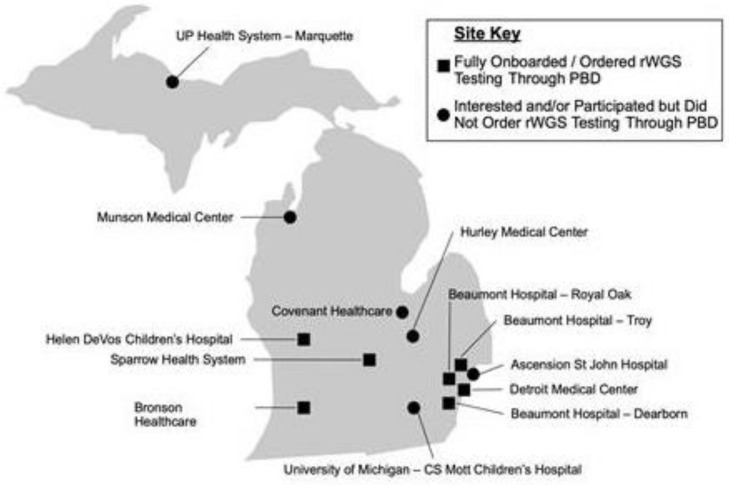
Map of Michigan Project and Baby Deer Hospital Sites.

**Table 1 children-10-00106-t001:** Inclusion/Exclusion Criteria for Project Baby Deer.

**Inclusion Criteria:** Inpatient at a MI project site <18 years old Meets one of the following criteria: ○ Admitted to a critical care unit OR ○ Admitted to another high-acuity in-patient unit and is suspected of having a genetic disease Meets one of the following criteria: Meets one of the following criteria: ○ Within 1 week of admission OR ○ Within 1 week of development of an abnormal response to standard therapy for an underlying condition
**Exclusion Criteria:** Patients whose clinical course is entirely explained by: Infection or sepsis with normal response to therapy Isolated prematurity Isolated unconjugated hyperbilirubinemia Hypoxic Ischemic Encephalopathy with clear precipitating event Previously confirmed genetic diagnosis that explains the clinical condition (e.g., have a positive genetic test Isolated Transient Neonatal Tachypnea Trauma Meconium aspiration
**Minimum genome-wide sequencing standards:** Given the data on the effects of turnaround time on economic impact and the clear yield of several specific tests, genomic sequencing could be sent to any laboratory that offered a complete test of all coding and non-coding sequences of clinically relevant genes, high resolution copy number (to single exon) for all OMIM morbid genes, mitochondrial (mt)DNA coverage, SMN1/2 copy number analysis and a consistent mean and median turnaround time of less than 3 days.

**Table 2 children-10-00106-t002:** Project Baby Deer Hospital Site Characteristics.

Onboarded Sites.	Site Type	Genetics on Site	NICU/PICU Admissions per Year (Estimated)	Date of Initial rWGS Test Sent
Beaumont Hospital- Dearborn	Community	No	NICU-759 No PICU	January 2021
Beaumont Hospital- Royal Oak	Community/Hybrid	No	NICU-1311 PICU-610	December 2020
Beaumont Hospital- Troy	Community	No	NICU-572 No PICU	June 2021
Bronson Methodist Hospital	Community/Hybrid	No	NICU-600 PICU-700	August 2020
Detroit Medical Center/ Hutzel Women’s Hospital	Free standing children’s hospital	Yes	NICU-1200 PICU-2100	May 2020
Helen DeVos Children’s Hospital	Free standing children’s hospital	Yes	NICU-1600 PICU-1319	November 2020
Sparrow Health System	Community/Hybrid	Yes	NICU-800 PICU-455	November 2020

**Table 3 children-10-00106-t003:** Diagnostic Rate and Change in Management Rate by Site.

Pilot Site	Children Who Received rWGS	Children Diagnosed (Diagnostic Rate)	Children Whose Care Changed (Change in Management Rate)
Beaumont Health, Dearborn	3	2 (67%)	2 (67%)
Beaumont Health, Royal Oak	6	0 (0%)	0 (0%)
Beaumont Health, Troy	1	1 (100%)	0 (0%)
Bronson Methodist Hospital	15	6 (40%)	7 (47%)
Children’s Hospital of Michigan	10	4 (40%)	1 (10%)
Helen DeVos Children’s Hospital	45	18 (40%)	11 (25%)
Sparrow Hospital	9	4 (44%)	3 (33%)
**Totals**	**89**	**35 (39%)**	**24 (27%)**

Changes in management include the addition, removal, or change in surgical interventions, procedures, medications, diet, length of stay, transplant.

**Table 4 children-10-00106-t004:** Economic Impact.

Total Cases in Cohort	89
Total Genomes Sequenced (proband + comparator)	140
Average Cost per Case *	$7563.86
Change in Management Surveys Completed	64
Midpoint of Estimated Range of Avoided Days	154.5
Avg Savings Per Avoided Inpatient Day	$4854.47
Gross Savings (n = 64)	$750,015.32
Net Savings (n = 64)	$265,928.53
**Net Savings Per Patient**	**$4155.13**

* Medicaid Fee Schedule: 0094U (rWGS proband, RCIGM)—$6278.06, 81,426 (rWGS comparator genome)—$2243.84.

## Data Availability

Deidentified data will be shared upon reasonable request directed to Caleb P. Bupp (caleb.bupp@spectrumhealth.org) from qualified investigators beginning 6 months and ending 5 years after study publication.

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
