# Peer review of "Breaking Barriers to Rapid Whole Genome Sequencing in Pediatrics: Michigan’s Project Baby Deer"

_children, 2023, doi:10.3390/children10010106_

Round 1
Reviewer 1 Report
I attached a .doc file with all the details

Author Response
We thank the reviewers for their thoughtful comments on our manuscript. The suggested revisions strengthened the paper. Our response to each of the comments is below the comment in red and denoted in track changes in the revised manuscript.
The authors report their own experience on the application of rapidWGS (rWGS) in pediatric hospital and, in particular in NICU e PICU. The project, named Michigan’s Project baby Deer (PBD) has been developed based on the previous experience of the Project baby Bear and reports the the positive results achieved and the difficulties that will still have to be faced
Major issue:
Although the manuscript is well written and it aims to illustrate the project without going into the details of the results obtained from the individual clinical cases, some issues should be reviewed and detailed.
- Line 72: authors cited the “new precise medicine technologies” as additional strategies to improve the clinical care. Could the authors take this speech back in the discussion and contextualize it?
Thank you for this suggestion. Further discussion around those technologies could include RNA sequencing, metabolomics and methylation analysis, among other topics. In order to allow focus for this manuscript, we have removed “other new precision medicine technologies” from the introduction section.
- Line 137: the authors talked about “phenotype-driven analysis”. I think that could be useful for readers to understand how the phenotypes has been considered in the present project (HPO terms? Clinical terms not standardized? Other?). Please add a paragraph or include this topic in the manuscript also highlight how the phenotype has been taken inot consideration for the diagnosis Additional information provided in this paragraph to clarify.
Moreover it is not clear why sequencing the Whole Genome may be an advantage over the Whole Exome sequencing. The authors should include e paragraph explaining this topic. About this, I also would sugguest to explain why in Table 1, authors talked about “genome wide sequencing” but reports only coding sequencing, CNV of single xons, mtDNA, SMN1/ CNV: is’t it a WES? This point is not clear in the whole manuscript. The authors agree that the discussion around genome versus exome is still ongoing. It is not the intention of this manuscript to debate this but to share a statewide workflow and results, which were done with WGS. Table 1 updated to reflect that non-coding sequence was included in WGS.
Minor issue:
- Table 1: Could the term “genetic diagnosis” modified in “genetic diseases” ? Change made
- Line 120: It is cited the Figure 1 and the table 2: Figure 1 is not present in the manuscript and Table 2 is too far from it citation. Please verify the figures and their numeration and move Table 2 after the paragraph 2.2 Figure 1 now included. Table 2 moved
- Line 123: typo rWSG to move in rWGS Correction made
- Line 152: please add the reference Reference 13 is listed here and included in the references, so unsure what reviewer is exactly suggesting.
- Paragraph from line 122 to line 134: this paragraph explain the economical aspects of the PBD. I would like to suggest to the authors to think about moving it in the “exploration” session. Paragraph relocated
- Table 3: I suggest to include how many singletons and trios were sequenced and the same for diagnosis. This could help readers understand whether the trios analysis could improve the diagnostic yield This information now added at the Section 3.1
- Table4: not rilevant. Please move in supplementary data Agree – change made
- paragraph 3.2: the economical impact and the saving cost should be calculated based on the disease of the patient. Please better explain if and how it is to considered an approximation. Please detail We agree, disease-specific analysis is appealing for this calculation. For this project, as it was done to particularly open up access to Medicaid authorization and payment, the team decided to solely calculate economics/savings on decreased length of stay. This was done partly for simplicity, partly to avoid any appearance of inflation of savings, and also to allow uniformity as multiple hospitals/centers participated.
- Paragraph 3.3: Is it possibile to distinguish between family who received the genetic diagnosis and those who have not? Please detail Feedback was received from both – clarification added to manuscript.
- Table 5: Are the “cost per case” e mean of what detailed in the legend? Please clarify That is correct. Cost is average taking 89 probands at $6278 and 51 comparators at $2243 then dividing that total by 89 cases. We have revised the label to “Average Cost per Case” in Table 5 to clarify this point.

Reviewer 2 Report
The authors show in this manuscript the development of Project Baby Deer (PBD) aimed at breaking the barriers of access to genomic technologies such as rWGS in critically ill neonatal and pediatric hospitalized patients in Michigan. I believe that this important initiative deserves to be published to the scientific and governmental community so that it can be maintained over time and extended to other states in the USA and the world, as well as to patients with other types of pathologies that require accurate diagnosis and/or personalized follow-up, which not only has a positive impact at the clinical level but also at the family, social, economic and political level.
I suggest making some adjustments to clarify some aspects obtained with the development of this project (PBD):
1. Expand a little the information regarding the genomic sequencing tests implemented in the different centers, both at a technological level (eg. platforms used, pipeline used, and so on) and analytically (eg. Consensus of data analysis guides used, and so on). It is important to reference under which standards or consensus the data derived from genomic tests are being carried out and used.
2. Lines 123-125: In line with the previous item, I recommend expanding this section: “Procedures for rWSG ordering, sample acquisition, and results reporting were performed according to previously published workflows, with a preference for trio testing when ultra-rapid whole genome sequencing was requested [12]."
3. The low number of geneticists in hospitals is notorious, which makes the clinical approach and genetic counseling of patients and relatives difficult. I suggest that in the results and discussion the information on the strategies that the project implemented in the face of this limitation be expanded and what perspectives are proposed to improve the roadmap in the face of this aspect.
In addition to the "Genomics 101" course, expand on other strategies that they have implemented, what barriers did they encounter and how did they face them throughout the project? In the future, what would be essential to improve to overcome this limitation?
4. Within the implementation phase, it is necessary to expand on the use of the information transmission portal. Did the project generate an information system aimed at storing all the information collected in the hospitals? How could it be planned to create a repository, database, web portal, and record there all the clinical, genomic, economic information, and so on, that contributes to the scientific, medical, social, and economic field?
5. It would be helpful to publish in supplementary material the impact evaluation survey used in the project.
6. In conclusion, I suggest synthesizing the future perspectives that remain with this project to sustain it over time and extend to other states of the country and the world.
I once again congratulate the authors for this initiative, it would be interesting if through different means of communication, events, and meetings they could disseminate their experiences gained and perspectives derived from this vital initiative for the improvement of public health in the world.
Author Response
We thank the reviewers for their thoughtful comments on our manuscript. The suggested revisions strengthened the paper. Our response to each of the comments is below the comment in red and denoted in track changes in the revised manuscript.
The authors show in this manuscript the development of Project Baby Deer (PBD) aimed at breaking the barriers of access to genomic technologies such as rWGS in critically ill neonatal and pediatric hospitalized patients in Michigan. I believe that this important initiative deserves to be published to the scientific and governmental community so that it can be maintained over time and extended to other states in the USA and the world, as well as to patients with other types of pathologies that require accurate diagnosis and/or personalized follow-up, which not only has a positive impact at the clinical level but also at the family, social, economic and political level.
- I suggest making some adjustments to clarify some aspects obtained with the development of this project (PBD):
- 1. Expand a little the information regarding the genomic sequencing tests implemented in the different centers, both at a technological level (eg. platforms used, pipeline used, and so on) and analytically (eg. Consensus of data analysis guides used, and so on). It is important to reference under which standards or consensus the data derived from genomic tests are being carried out and used. Further information now added to Section 2.3 in Implementation
- 2. Lines 123-125: In line with the previous item, I recommend expanding this section: “Procedures for rWSG ordering, sample acquisition, and results reporting were performed according to previously published workflows, with a preference for trio testing when ultra-rapid whole genome sequencing was requested [12]." Additional information added. Please note, this section was shifted to 2.1 in the manuscript per recommendation from Reviewer 1.
- 3. The low number of geneticists in hospitals is notorious, which makes the clinical approach and genetic counseling of patients and relatives difficult. I suggest that in the results and discussion the information on the strategies that the project implemented in the face of this limitation be expanded and what perspectives are proposed to improve the roadmap in the face of this aspect. We completely agree. Comments are included in implementation section about monthly case review being offered for all participating and non-participating sites to bring awareness to genetics and non-genetics providers. Additional information regarding this suggestion now included in discussion.
- In addition to the "Genomics 101" course, expand on other strategies that they have implemented, what barriers did they encounter and how did they face them throughout the project? In the future, what would be essential to improve to overcome this limitation? Additional commentary around education done at Michigan conferences/meetings is now provided in Discussion.
- 4. Within the implementation phase, it is necessary to expand on the use of the information transmission portal. Did the project generate an information system aimed at storing all the information collected in the hospitals? How could it be planned to create a repository, database, web portal, and record there all the clinical, genomic, economic information, and so on, that contributes to the scientific, medical, social, and economic field? No information portal was created or used for this study. Suggestion of this now included in last paragraph of Discussion.
- 5. It would be helpful to publish in supplementary material the impact evaluation survey used in the project. That assessment now included as Supplementary Figure 1 at end of manuscript
- 6. In conclusion, I suggest synthesizing the future perspectives that remain with this project to sustain it over time and extend to other states of the country and the world. Additional summary added to Conclusions.
- I once again congratulate the authors for this initiative, it would be interesting if through different means of communication, events, and meetings they could disseminate their experiences gained and perspectives derived from this vital initiative for the improvement of public health in the world.
Thank you!
